# Enhancing Chat Language Models by Scaling High-quality Instructional Conversations

**Ning Ding**[1*] , **Yulin Chen**[2,3*], **Bokai Xu**[4], **Yujia Qin**[2,3], **Shengding Hu**[2,3]
**Zhiyuan Liu**[2,3†] , **Maosong Sun**[2,3†], **Bowen Zhou**[1†]
[1]Department of Electronic Engineering,Tsinghua University
[2]Department of Computer Science and Technology, Tsinghua University
[3] BNRIST, IAI, Tsinghua University, [4]The Chinese University of Hong Kong, Shenzhen

## Abstract

Fine-tuning on instruction data has been widely validated as an effective practice for implementing chat language models like ChatGPT. Scaling the diversity and quality of such data, although straightforward, stands a great chance of leading to improved performance. This paper aims to push the upper bound of open-source models further. We first provide a systematically designed, diverse, informative, large-scale dataset of instructional conversations, UltraChat, which does not involve human queries. Our objective is to capture the breadth of interactions between a human user and an AI assistant and employs a comprehensive framework to generate multi-turn conversation iteratively. UltraChat contains 1.5 million high-quality multi-turn dialogues and covers a wide range of topics and instructions. Our statistical analysis of UltraChat reveals its superiority in various key metrics, including scale, average length, diversity, coherence, etc., solidifying its position as a leading open-source dataset. Building upon UltraChat, we fine-tune a LLaMA model to create a powerful conversational model, UltraLM. Our evaluations indicate that UltraLM consistently outperforms other open-source models, including WizardLM and Vicuna, the previously recognized state-of-the-art open-source models.

## 1 Introduction

Large language models (Bommasani et al., 2021; Han et al., 2021; Chowdhery et al., 2022) (LLMs) have demonstrated exceptional generalization capability on a variety of language-related tasks. Notably, ChatGPT (OpenAI, 2022), an optimized version of GPT-3 (Brown et al., 2020) for conversation, along with GPT-4 (OpenAI, 2023), takes the user experience to another level via excelling in comprehending and generating responses in a natural and interactive manner. The introduction of

| Model | Score |
|---|---|
| Dolly-v2 (Conover et al., 2023) | 4.04 ± 2.34 |
| MPT-Chat (Mosaic, 2023) | 6.67 ± 2.88 |
| OpenAssistant (Köpf et al., 2023) | 7.65 ± 2.15 |
| Alpaca (Taori et al., 2023) | 8.04 ± 2.05 |
| Koala (Geng et al., 2023) | 8.23 ± 1.99 |
| Baize (Xu et al., 2023b) | 8.50 ± 1.34 |
| Vicuna (Chiang et al., 2023) | 8.78 ± 1.55 |
| Wizard-LM (Xu et al., 2023a) | 8.95 ± 1.44 |
| UltraLM (ours) | **9.00 ± 1.33** |

Table 1: Average scores (1-10) across different open-source models and UltraLM. The evaluation is conducted on our curated evaluation set with GPT-4. Evaluation prompts can be found in Appendix C.

ChatGPT has spurred a surge in the adoption and implementation of general chat language models.

In addition to competing models developed by large corporations such as Bard[1] and Claude[2], the open-source community is actively engaged in training similar models, aiming to democratize access to AI technology. Notable examples in this regard include Alpaca (Taori et al., 2023), Vicuna (Chiang et al., 2023), Koala (Geng et al., 2023), Baize (Xu et al., 2023b), and Belle (Ji et al., 2023), etc., demonstrating promising performance. Experimental evidence strongly suggests that chat language models can be effectively trained through instruction fine-tuning (Wei et al., 2021; Sanh et al., 2021), and they also indicate that many data-efficient (Zhou et al., 2023) or computing-efficient (Hu et al., 2021; Ding et al., 2023) methods can be applied. This paper, in another way, focuses more on the "final one mile" of chat language models, as evidence shows that **the journey from 0 to 60 is easy, whereas progressing from 60 to 100 becomes exceedingly challenging**. For instance, researchers have shown that by utilizing a small,

---

\* equal contributions
† Corresponding authors

[1]https://bard.google.com/
[2]https://www.anthropic.com/index/introducing-claude

thoughtfully curated set of instructions, it is possible to train a model with satisfactory instruction-following capabilities. However, these approaches have yet to produce models that surpass the performance of Vicuna, the current leading open-source model, let alone outperform ChatGPT and GPT-4.

This paper believes that the most straightforward way, that is, the quality and diversity of training data, play a vital role in further improving the performance of chat language models. In other words, leveraging higher quality and more diverse data can yield better outcomes. To this end, we present UltraChat, a million-scale multi-turn instructional conversation data, to facilitate the construction of more powerful chat language models. UltraChat is carefully designed to capture the breadth of interactions that a human might have with an AI assistant. Specifically, we do not use specific tasks like question-answering or summarization to construct the data, but curate three sectors: Questions about the World, Creation and Writing, and Assistance on Existing Materials. Then we employ meta-information, in-context expansion, and iterative prompting to scale up the number of instructions. To construct informative and realistic multi-turn conversations, two separate ChatGPT Turbo APIs are adopted in the conversation generation, where one plays the role of the user to generate queries, and the other generates the response. We instruct the user model with carefully designed prompts to mimic human user behavior and call the two APIs iteratively.

We fine-tune a LLaMA-13B model on Ultra-Chat to produce UltraLM and compare the model to a wide range of baselines, especially the open-source ones. The evaluation shows that our model could consistently outperform other models. As reported in Table 1, UltraLM achieves the highest performance scores that are independently assessed by GPT-4. Further evaluation results on challenging benchmarks and preference study with GPT-4 on various evaluation sets also show that UltraLM could surpass all other open-source models.

## 2 Related Work

**Instruction Tuning.** Recent works demonstrate LLMs' powerful capabilities in following human instructions. Wei et al. (2021) pioneered to fine-tune T5 (Raffel et al., 2020) on 60 NLP datasets verbalized with natural language instruction templates, i.e., *instruction tuning*. The fine-tuned model ex-

hibits a strong ability in instruction understanding and generalizes well to unseen instructions (Sanh et al., 2021; Ouyang et al., 2022). Later, Longpre et al. (2023) show the benefits of scaling the number of tasks in out-of-distribution generalization. Wei et al. (2021) also conclude that the success of instruction tuning depends on the quality of the dataset and the design of prompts. To further regulate the tuned model's behavior, Ouyang et al. (2022); Schulman et al. (2017) propose to employ reinforcement learning to align model behaviors with human preferences. This technique combined with instruction tuning can further boost the model performance and has been successfully applied to LLMs such as ChatGPT.

**Data Augmentation with LLMs.** Collecting large-scale human-annotated instructions and their responses is time-consuming and labor-intensive. Alternatively, a more cost-effective and feasible approach to gathering top-notch data involves sampling from LLMs that have been well-tuned, e.g., ChatGPT and GPT-3.5. Recently, there is a surge of interest in distilling these powerful LLMs for data augmentation. For instance, using the technique of Self-Instruct (Wang et al., 2022), Alpaca (Taori et al., 2023) generate 52k high-quality instruction-response pairs based on seed tasks by "distilling" Text-Davinci-003. The trained model performs almost on par with Text-Davinci-003. The success of Alpaca boosts numerous later efforts on data augmentation with LLMs, such as code-alpaca (Chaudhary, 2023), alpaca-cot (Si et al., 2023), GPT4ALL (Anand et al., 2023), ShareGPT (Domeccleston, 2023), Dolly-v2 (Conover et al., 2023), BELLE (Ji et al., 2023), Vicuna (Chiang et al., 2023), Koala (Geng et al., 2023), Baize (Xu et al., 2023b), etc. It is shown that increasing the scale of data could constantly improve the model performance. Besides, prompt engineering also affects data quality. CAMEL (Li et al., 2023a) design a multi-agent role-play environment for LLMs to simulate real human conversations.

## 3 Data Construction

LLMs are believed to be better annotators than human-being in many scenarios (Gilardi et al., 2023). However, employing LLMs such as Chat-GPT directly for generating multi-turn conversations may yield satisfactory but less informative results, as it cannot enjoy the benefit of reinforce-

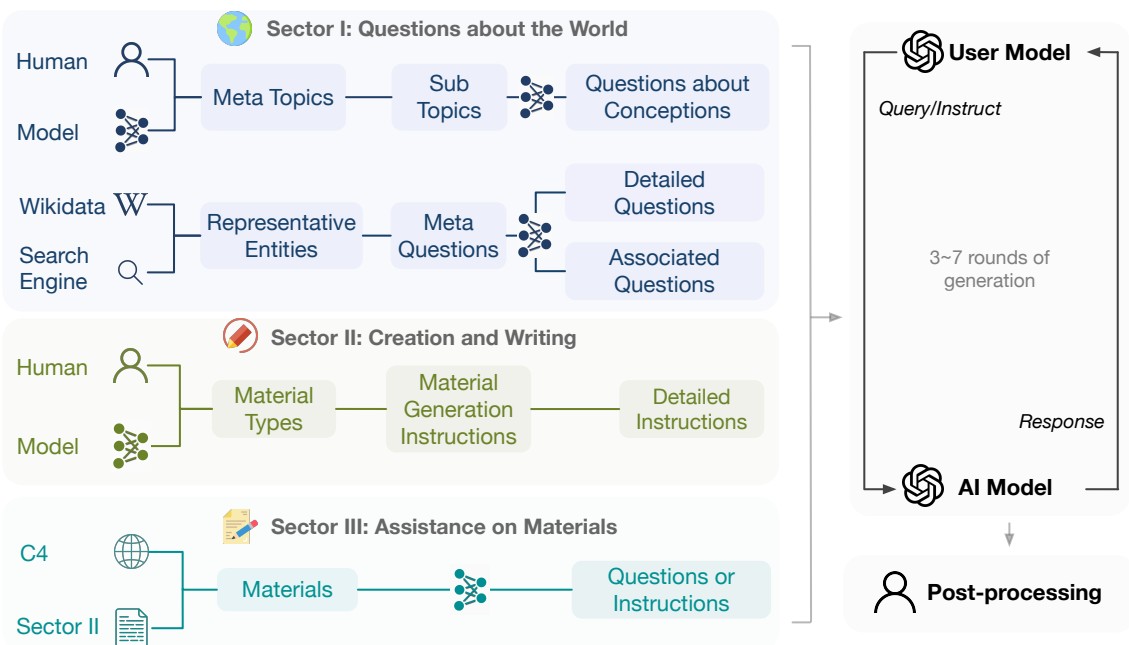

Figure 1: Construction process of UltraChat. The three sectors of data are derived from different meta-information.

ment learning with human feedback (RLHF) in the alignment process. Table 12 in Appendix A shows a comparison of directly generated multi-turn dialogue and a case in UltraChat with the same opening line. Two key points can be derived to ensure the quality of the data: (1) An *opening line* determines the topic of the dialogue. Opening lines should be highly diverse and encompass any task that a human user may request a chat model to perform. (2) A *user* determines the plot of the dialogue, and the output should be tailored to the current topic with diverse language styles and requests.

Therefore, unlike traditional task-specific datasets, to construct a comprehensive open-domain instructional chat dataset, the design of data collection schema is crucial to capturing the breadth of interactions and ensuring data quality. UltraChat aims to cover a tremendous range of conversation data with a carefully designed tripartite schema: *Questions about the World*, *Creation and Writing*, and *Assistance on Existing Materials*. While the core of ensuring data diversity mainly depends on opening line diversity, we will first introduce the idea behind the sector design and then focus on specific measures to obtain a diverse set of opening lines and how to prompt the user properly.

## 3.1 Questions about the World

The first sector focuses on querying existing information in the world, including concepts, objects, and entities that exist in the real world. This is at the core of human-AI interaction, as users often rely on AI assistants to provide quick and accurate answers to their questions.

Our approach to gathering data for this sector involves two perspectives: one centered around topics and concepts, and the other around real-world entities. Initially, we request ChatGPT to generate 30 comprehensive topics that encompass various aspects of our daily lives, as shown in Table 2. Subsequently, we delve deeper into each topic by generating 30 to 50 subtopics or related concepts. Finally, we generate 10 different questions for each subtopic or concept and additionally request Chat-GPT to generate 10 more questions based on each original question. The other source of data comes from real-world objects, which are derived from Wikidata[3] entities. These entities are further refined by considering their frequencies in Wikipedia[4] articles, specifically focusing on the 10,000 most frequently occurring entities. For each entity, we create 5 meta-questions, followed by 10 more specific questions and 20 extended questions. The extended questions aim to maintain some similarity to the original question while exploring distinct objects or topics. To create a dialogue, we filter and sample approximately 500,000 questions as opening lines. During the construction of each dialogue, we provide the user model with carefully crafted prompts that explicitly ask the model to respond

---

[3] https://www.wikidata.org/
[4] https://www.wikipedia.org/

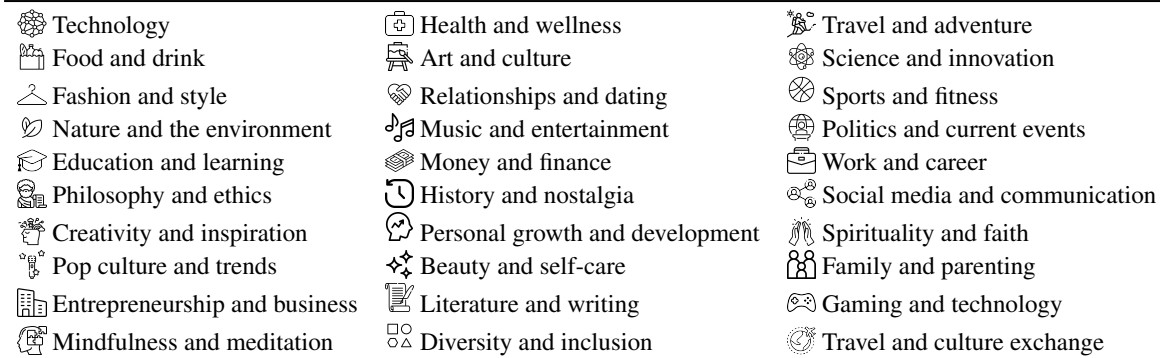

| ⚙ Technology | ⊕ Health and wellness | ⛷ Travel and adventure |
|---|---|---|
| 🏪 Food and drink | 🏛 Art and culture | ⚛ Science and innovation |
| 👝 Fashion and style | 🔮 Relationships and dating | ⊗ Sports and fitness |
| 🍃 Nature and the environment | 🎶 Music and entertainment | 🌐 Politics and current events |
| 🎓 Education and learning | 💸 Money and finance | 💼 Work and career |
| 📇 Philosophy and ethics | 🕐 History and nostalgia | 💬 Social media and communication |
| 🌱 Creativity and inspiration | 🗺 Personal growth and development | 👣 Spirituality and faith |
| 🏆 Pop culture and trends | ✨ Beauty and self-care | 👥 Family and parenting |
| 🏢 Entrepreneurship and business | 📝 Literature and writing | 🎮 Gaming and technology |
| 🧘 Mindfulness and meditation | 🔳 Diversity and inclusion | ⌚ Travel and culture exchange |

Table 2: 30 meta-concepts used to generate the first sector of UltraChat data.

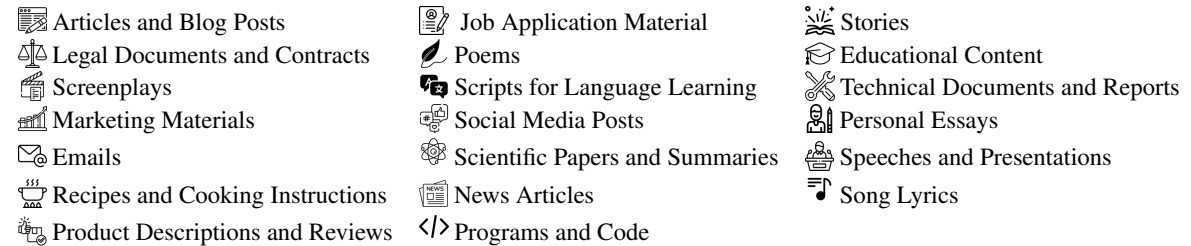

| 📰 Articles and Blog Posts | 📋 Job Application Material | 🎎 Stories |
|---|---|---|
| ⚖ Legal Documents and Contracts | 🖊 Poems | 📘 Educational Content |
| 🏢 Screenplays | 🎴 Scripts for Language Learning | 🔧 Technical Documents and Reports |
| 📈 Marketing Materials | 📱 Social Media Posts | 📑 Personal Essays |
| 📧 Emails | ⚛ Scientific Papers and Summaries | 🎤 Speeches and Presentations |
| 🍲 Recipes and Cooking Instructions | 📰 News Articles | 🎵 Song Lyrics |
| 🛍 Product Descriptions and Reviews | </> Programs and Code | |

Table 3: 20 types of text materials used for sector 2 and 3 UltraChat construction.

concisely and meaningfully, taking into account the context of the ongoing dialogue history.

## 3.2 Creation and Writing

The second part is concerned with the creation of new information with human-input conditions, ranging from writing emails to crafting stories and plays. This process reflects the AI's capacity to engage in original content generation alongside users and demonstrates the role of AI assistants as collaborative partners in a creative environment.

We first project all creations as text materials, and further categorize them into 20 different types as in Table 3. Then a ChatGPT model is employed to produce a diverse range of instructions for each type of writing, approximately 80% of which are further refined by ChatGPT model to generate more detailed instructions. These instructions serve as opening lines for dialogue generation. Throughout the generation process, the user prompt constantly reinforces the primary objective of the conversation, which is to generate and refine a piece of writing. This serves to ensure that the behavior of the user model remains focused and aligned with the intended purpose.

## 3.3 Assistance on Existing Materials

The third sector mainly addresses the modification of existing information, encompassing various tasks including rewriting, translation, summa-

rization, and question-answering, etc. Modifying existing materials is a crucial aspect of human-AI interaction, as it allows the AI assistant to actively engage with the user's input, transforming it in various ways as instructed by the user.

We begin by gathering text pieces from the C4 corpus[5]. Each piece within the C4 corpus is associated with a source URL. To ensure a diverse range of text content and styles, we adopt the 20 material types outlined in the previous section and manually curate keywords for each type. Additionally, we classify the text in the corpus by matching the keywords in the corresponding URL. In total, we collect 100,000 text pieces from the C4 corpus, and for each piece, we prompt ChatGPT to generate five distinct instructions. We use a manually designed template to combine text and instructions, as depicted in Figure 4 in the appendix. Ultimately, the concatenated set of 500,000 pieces serves as the opening lines for the generated dialogues.

## 3.4 User Simulation and Refinement

Maintaining the desired behavior of the user model is crucial for achieving successful automatic dialogue generation. It has been observed that when the user model is solely provided with the current dialogue history, it tends to assume the role of an AI assistant. This "role confounding" situation

---

[5] https://commoncrawl.org/

| Dataset | #Dialogue | Avg. #Turns | Avg. Dialog Length (by token) | Avg. Utt. Length (by token) | Lexical Diversity ($\uparrow$) | Topic Diversity ($\downarrow$) | Coherence ($\uparrow$) | User Simulation |
|---|---|---|---|---|---|---|---|---|
| Self-Instruct | 82,439 | 1 | 69.8 | 29.2 | 24.9 | 0.733 | - | No |
| Stanford Alpaca | 52,002 | 1 | 91.1 | 64.5 | 42.8 | 0.727 | - | No |
| SODA | **1,486,869** | 3.6 | 231.8 | 22.5 | 38.6 | 0.797 | 8.48 | No |
| GPT-4-LLM | 61,002 | 1 | 179.6 | 142.9 | 48.9 | 0.721 | - | No |
| BELLE | 1,436,679 | 1 | 102.3 | 63.3 | 35.9 | 0.771 | - | No |
| Baize | 210,311 | 3.1 | 293.9 | 52.8 | 67.1 | 0.751 | **9.06** | Yes |
| GPT4ALL | 711,126 | 1 | 597.7 | **318.9** | 62.7 | **0.692** | - | No |
| UltraChat | 1,468,352 | **3.8** | **1467.4** | 309.3 | **74.3** | 0.702 | 9.06 | Yes |

Table 4: Statistics of existing instruction datasets. Lexical diversity is calculated by averaging the MTLD score (Mc-Carthy and Jarvis, 2010) over each utterance with LexicalRichness[6]. 10000 samples are randomly drawn from each dataset for topic diversity and coherence measurement. Topic diversity is measured by averaging the cosine distance between each pair of data with OpenAI embedding API. Coherence is scored by ChatGPT on a scale of 1-10.

can significantly deteriorate the coherence of the multi-turn conversation. To address this, in addition to presenting the dialogue history, we include prompts explicitly instructing the model to adopt various user personalities. In Sector 2, a prompt is employed to remind the model of the primary purpose of the dialogue, thereby promoting a more natural conversation flow. Once the data generation process is complete, a further filtration step is performed to ensure overall data quality. We also exclude excessively polite statements to enhance the realism of user responses.

## 4 Data Analysis

### 4.1 Statistical Analysis

We conduct a statistical analysis of UltraChat and several other instruction datasets, as shown in Table 4. UltraChat stands out in terms of its scale, being one of the largest publicly available datasets. Moreover, it exhibits the highest average number of turns and the longest average length per instance of data. While SODA (Kim et al., 2023) also has many rounds, it is primarily composed of conceptual banter rather than instructional content. Additionally, the average number of tokens per dialogue in SODA is 231.8, whereas UltraChat boasts a remarkable 1467.4 tokens. To evaluate diversity, we measure both lexical diversity and topic diversity. UltraChat outperforms previous datasets in terms of lexical diversity. However, in terms of topic diversity, UltraChat falls slightly short compared to GPT4ALL (Anand et al., 2023) but still surpasses other datasets significantly. This may be attributed to the regularized embeddings resulting from a large number of tokens in each dialogue. We also conduct coherence evaluation with ChatGPT for multi-turn datasets. Notably, UltraChat and Baize data rank the highest in terms of coherence.

### 4.2 Human Assessment

**Setup.** To better evaluate the constructed data quality, we also conduct human assessment for UltraChat. Due to the difficulty of evaluation of multi-turn dialogue and the resulting formidable cost, we sample 500 representative dialogues for human evaluation, among which 300 are from UltraChat sector 1, 100 from sector 2 and sector 3 respectively. For each round of conversation, we ask the annotators to score the assistant's response on Helpfulness, Honesty, and Harmlessness (3H) principles (Askell et al., 2021). We also devise Coherence and Consistency criteria for the overall multi-turn dialogue quality evaluation. Coherence evaluates whether the dialogue flows logically and coherently, for which the annotators evaluate both the user's response and the assistant's response. Consistency means the assistant's responses do not contradict each other within the same dialogue. For example, it is inconsistent if the assistant asserts one specific event occurred in 1911 in the first round of conversation but mentions it as a 1901 event in the next round. Each metric is scored with 0, 0.5 or 1, where higher score means better quality. Therefore, for a K-round dialogue, we have $3K+2$ metric annotations.

**Annotation.** Each dialogue is annotated independently by two well-trained annotators, and the score is averaged across two annotators. Meanwhile, due to the difficulty in identifying the hallucination problem, we allow the annotators to skip the dialogues that require expert knowledge or whose validity is hard to check. Altogether, we collect 14560 valid annotations in terms of metrics for both single-round and multi-round, and the Cohen's kappa coefficient is 0.358. The average time to annotate one dialogue is 10 minutes.

**Results.** As shown in Table 5, the dataset scores high on all metrics, showing the effectiveness of the construction process. It is worth noting that the dataset is almost free from harmful content like hate speech and discrimination. Furthermore, we observe two trade-offs between data quality. The first is between helpfulness and honesty. While the honesty score is pretty high, helpfulness is compromised. It is because some user queries are out of the LLM ability scope (e.g., ask the LLM to compose a song). Under such circumstances, the LLM refuses to answer the question with an explanation of incapability, which is reasonable and expected but less helpful. The phenomenon is particularly prominent in part 3, as the simulated user often asks for information not existent in the text material. The second trade-off is between control and dialogue naturalness. While sector 2 and sector 3 data are constructed with more guidance and control when prompting for user simulation, they have clearly lower quality in coherence and consistency than sector 1.

| Data | Helpful | Honest | Harmless | Coherent | Consistent |
|---|---|---|---|---|---|
| **Sector 1** | 0.971 | 0.996 | 1.000 | 0.996 | 0.995 |
| **Sector 2** | 0.978 | 0.986 | 1.000 | 0.982 | 0.977 |
| **Sector 3** | 0.893 | 0.983 | 1.000 | 0.964 | 0.981 |
| **Overall** | 0.960 | 0.992 | 1.000 | 0.987 | 0.988 |

Table 5: Human assessment results on 500 dialogues sampled from UltraChat.

# 5 Experiments

We developed UltraLM, an enhanced variant of the LLaMA-13B (Touvron et al., 2023) model, by training it on the UltraChat dataset. To improve the model's comprehension of dialogue context, we break down each dialogue into smaller sequences, limiting them to a maximum length of 2048 tokens. During the training process, we only calculate the loss for the model's responses. This approach ensured that the model had access to the relevant information from earlier parts of the conversation, enabling a more comprehensive understanding of the ongoing dialogue. By incorporating the preceding context, UltraLM was equipped to generate more contextually appropriate and coherent responses. We use standard cross-entropy loss to finetune the model. The model is trained with 128 A100 GPUs and the total batch size is 512.

The evaluation of the trained model is conducted in two folds. We first evaluate UltraLM on traditional benchmark datasets to delineate the knowledge scope and the multiple abilities of the language model. To better demonstrate the chat ability of language models, an automatic response quality evaluation is performed to showcase the model's proficiency in delivering accurate and informative content during chat interactions. Note that UltraLM is solely trained on UltraChat dataset without further finetuning on task specific datasets.

## 5.1 Experimental Setup

**Baselines.**[7] We mainly compare with other open-source instruction-tuned language models based on LLaMA (Touvron et al., 2023) and Pythia (Biderman et al., 2023) backbone model. The main baseline models include Alpaca (Taori et al., 2023), Vicuna (Chiang et al., 2023), Koala (Geng et al., 2023), Dolly (Conover et al., 2023), OpenAssistant (Köpf et al., 2023), and WizardLM (Xu et al., 2023a). The parameter size of the baselines ranges from 7B to 13B, which is comparable to UltraLM. Our evaluation also includes other chat language models like ChatGPT (OpenAI, 2022), MPT (Mosaic, 2023), and Baize (Xu et al., 2023b). A detailed description of the main baselines can be found in Appendix A.1.

**Datasets.** For *benchmark evaluation*, we choose four datasets: ARC-Challenge (Clark et al., 2018), HellaSwag (Zellers et al., 2019), MMLU (Hendrycks et al., 2021), and TruthfulQA (Lin et al., 2021), evaluating commonsense knowledge, professional knowledge and complex reasoning and understanding abilities. Each benchmark is constructed as multiple choice questions and therefore metrics are readily computable. The four datasets prove to be challenging even for the best-performing language models like ChatGPT.

For *response quality evaluation*, we use 3 datasets. We first create an evaluation set by ourselves. The curated set encompasses the Vicuna benchmark as well as an additional 300 questions and instructions generated by GPT-4. The questions/instructions covered a wide range of topics, including commonsense, world knowledge, professional knowledge (specifically physics and biology), mathematics, response generation, and writing tasks on different levels of difficulty. Apart from the curated set, we also adopt AlpacaEval (Li et al., 2023b), a widely acknowledged

---

[7]Some baselines used in our experiments are continuously updated. All the results in this section are from the latest versions of these baselines before June 23, 2023

| Model | ARC-Challenge | | HellaSwag | | MMLU | | TruthfulQA | | Overall |
| | Acc. | Acc. norm. | Acc. | Acc. norm. | Weighted | Unweighted | mc1 | mc2 | **Average** |
|---|---|---|---|---|---|---|---|---|---|
| Dolly-12B | 38.23 | 42.24 | 54.59 | 72.6 | 31.52 | 31.70 | 20.69 | 34.06 | 45.15 |
| OpenAssistant-12B | 41.38 | 45.90 | 52.51 | 70.04 | 29.77 | 30.29 | 24.60 | 39.29 | 46.38 |
| MPT-7B | 43.00 | 46.67 | 57.13 | 75.50 | 37.76 | 38.33 | 27.17 | 40.16 | 50.17 |
| Alpaca-7B | 49.74 | 52.65 | 58.05 | 76.91 | 42.47 | 42.90 | 25.83 | 39.55 | 53.00 |
| LLaMA-13B | 53.16 | 56.40 | 60.64 | 80.87 | 46.05 | 46.74 | 25.83 | 39.90 | 55.98 |
| Baize-13B | 55.55 | 57.94 | 59.96 | 80.36 | 48.13 | 49.03 | 32.93 | 47.43 | 58.69 |
| Koala-13B | 49.83 | 52.90 | 57.60 | 77.54 | 46.75 | 48.01 | 34.64 | 50.09 | 57.14 |
| Vicuna-13B | 51.71 | 52.90 | 60.03 | 80.12 | 50.15 | 50.45 | 35.74 | 51.82 | 58.83 |
| WizardLM-13B | 55.12 | 57.08 | 60.93 | 80.91 | **51.69** | **52.25** | 35.37 | 50.53 | 60.19 |
| LLaMA-65B | 59.22 | 63.31 | 66.40 | 86.05 | 62.29 | 62.97 | 27.91 | 42.55 | 63.72 |
| UltraLM-13B | **57.25** | **59.22** | **61.32** | **81.49** | 50.45 | 51.10 | **36.72** | **52.00** | **60.95** |

Table 6: The evaluation results on 4 challenging benchmark datasets. All evaluation and metric calculations follow EleutherAI's lm-evaluation-harness (Gao et al., 2021). Both weighted and unweighted mean accuracy are reported for MMLU as there are 57 tasks. The overall average metric is obtained by averaging the second column data for each benchmark dataset. More details about metric calculation can be found in Appendix A.3.

open-source evaluation set and leaderboard specifically designed for evaluating LLMs. The leaderboard is created based on the win-rate against Text-Davinci-003 automatically evaluated by GPT-4. To further compare with the state-of-the-art model WizardLM (Xu et al., 2023a), comparison result obtained with GPT-4 on the released Evol-Instruct (Xu et al., 2023a) test set is also reported. Further benchmark dataset and implementation details can be found in Appendix A.2 and A.3.

## 5.2 Benchmark Evaluation

As shown in Table 6, with pure instruction-tuning on the UltraChat dataset, UltraLM significantly improves over LLaMA-13B and achieves the best overall performance across four benchmarks. It is worth noting that UltraLM overtakes the current state-of-the-art model by nearly 2% on ARC-Challenge and TruthfulQA. It shows that UltraLM is equipped with both broad and profound comprehension of the world and commonsense knowledge. The improvement could be attributed to the systematic and comprehensive data construction process of UltraChat sector 1, which effectively extends and deepens the discussion about world knowledge in automatic conversation generation. Meanwhile, the comparative inferiority in MMLU hints on the lack of professional knowledge in specific fields. It suggests the need for more advanced data generation techniques to build a specialized expert language model. As for HellaSwag, we notice that all models have only marginal improvement compared to LLaMA-13B. It is probably because HellaSwag is formatted as an in-text completion task instead

of a straightforward text completion, and therefore benefits little from instruction tuning.

## 5.3 Response Quality Evaluation

**Response Comparison.** For our curated evaluation set, we conduct pairwise evaluation between UltraLM and each baseline model with GPT-4. Our evaluation prompt is designed to prioritize correctness over other factors such as informativeness. To mitigate the influence of presentation order of responses, we randomly determine the order of the responses for each question. Finally, we count the number of Win/Tie/Lose times against each baseline model, and the result is presented in Figure 2. UltraLM demonstrates superior performance compared to every open-source model, exhibiting an impressive winning rate of up to 98%. It is worth noting that UltraLM also outperforms Vicuna with 9% higher winning rate.

**Independent Scoring.** Given the instability of pairwise comparison, we also conduct independent quality scoring with GPT-4, as presented in Table 7. Notably, our model demonstrates superior performance compared to all the open-source counterparts by a significant margin in terms of overall scores. This breakdown also provides insights into the performance of each model on specific types of questions and instructions. Generally, all models perform better on simpler questions pertaining to commonsense knowledge and general world understanding. However, more complex tasks that involve reasoning and creative writing proves to be challenging for most models. Interestingly,

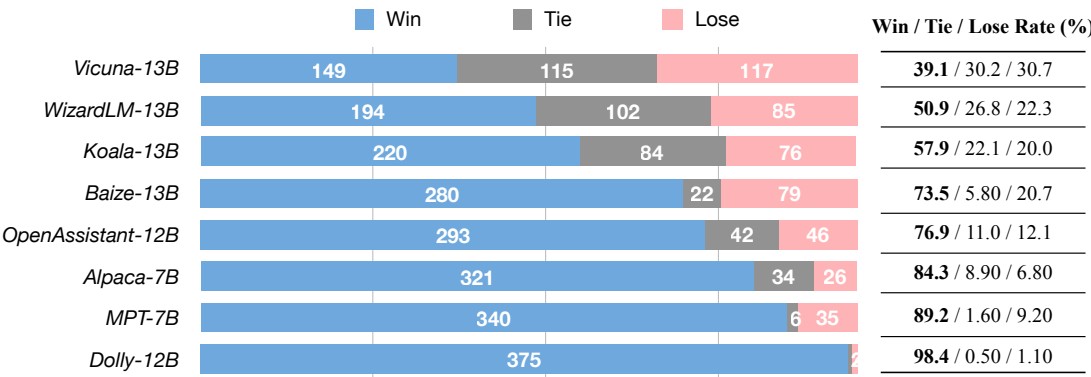

Figure 2: Response comparison of UltraLM with other baselines on the curated evaluation set, evaluated by GPT-4.

| Model | Vicuna Set | Commonsense | | World Knowledge | | Professional Knowledge | | Ability | | Writing | Overall |
|---|---|---|---|---|---|---|---|---|---|---|---|
| | | Easy | Moderate | Easy | Difficult | Physics | Biology | Math | Reasoning | | |
| Dolly-12B | 4.75 | 3.50 | 3.93 | 3.10 | 4.13 | 4.87 | 5.47 | 2.70 | 2.03 | 4.51 | 4.04 |
| MPT-7B | 7.25 | 5.57 | 8.20 | 5.53 | 5.87 | 7.83 | 8.40 | 5.97 | 3.97 | 7.25 | 6.67 |
| LLaMA-13B | 6.85 | 8.43 | 8.43 | 8.57 | 8.50 | 7.90 | 8.40 | 6.97 | 6.73 | 6.79 | 7.49 |
| OpenAssistant-12B | 7.88 | 8.13 | 7.80 | 9.13 | 7.50 | 8.10 | 8.20 | 6.57 | 5.17 | 7.75 | 7.65 |
| Alpaca-7B | 7.58 | 9.17 | 8.83 | 9.30 | 8.73 | 8.13 | 8.80 | 6.70 | 6.27 | 8.05 | 8.04 |
| Koala-13B | 8.00 | 9.20 | 9.07 | 9.00 | 8.93 | 8.53 | 9.07 | 7.33 | 5.30 | 8.40 | 8.23 |
| Baize-13B | 8.40 | 9.03 | 9.10 | 9.03 | 8.93 | 8.83 | 8.80 | 7.43 | 8.30 | 8.10 | 8.50 |
| Vicuna-13B | 8.48 | 9.67 | 9.50 | 9.37 | 9.30 | 9.23 | 9.33 | 8.07 | 6.90 | 8.76 | 8.78 |
| WizardLM-13B | 8.55 | **9.70** | 9.30 | 9.57 | 9.50 | **9.27** | 9.53 | 8.27 | 8.20 | 8.83 | 8.95 |
| ChatGPT | **9.15** | 9.67 | **9.60** | **9.80** | **9.60** | 9.17 | 9.73 | **9.33** | **9.13** | **8.95** | **9.31** |
| UltraLM-13B | 8.98 | **9.70** | 9.50 | 9.47 | 9.40 | **9.27** | **9.87** | 8.77 | 6.80 | 8.90 | 9.00 |

Table 7: The independent overall scoring and segment scoring of each model on the curated evaluation set, on a scale of 1 to 10. **Bold** indicates the best score and underlined indicates the second best.

Alpaca, despite having only 7 billion parameters, performs comparatively well with larger models on questions related to commonsense and world knowledge. Meanwhile, Dolly and OpenAssistant, which are based on Pythia (Biderman et al., 2023), display inferior performance compared to models based on LLaMA of similar or even smaller sizes. This observation highlights the significance of the underlying backbone language model.

| Model | Win Rate (%) | Standard Error |
|---|---|---|
| GPT-4 | 95.28 | 0.72 |
| Claude | 88.39 | 1.11 |
| ChatGPT | 86.09 | 1.21 |
| **UltraLM-13B** | **76.09** | 1.50 |
| WizardLM-13B | 75.31 | 1.51 |
| Guanaco-65B | 71.80 | 1.59 |
| Vicuna-13B | 70.43 | 1.61 |
| Oasst-RLHF-33B | 66.52 | 1.66 |
| Text Davinci 003 | 50.00 | 0.00 |
| Falcon-40B-instruct | 45.71 | 1.75 |
| Alpaca-7B | 26.46 | 1.54 |

Table 8: Win rates against Text-Davinci-003 on AlpacaEval leaderboard. GPT-4 is used for evaluation following the official implementation (Li et al., 2023b).

**AlpacaEval.** As shown in Table 8, UltraLM outperforms existing models in terms of win rate on the current AlpacaEval leaderboard and ranks 4th just below ChatGPT. This observation further testifies the response quality of UltraLM and is in line with results on our curated dataset. Furthermore, a significant gap is evident between UltraLM and ChatGPT performance, highlighting the substantial effort needed for open-source models to match the capabilities of ChatGPT.

**Evol-Instruct Evaluation.** Figure 3 shows the automatic comparison results against WizardLM-13B on Evol-Instruct test set. UltraLM overtakes WizardLM on most types of questions, with up to 29% increase in scores. It is important to acknowledge that WizardLM-13B is trained on the Evol-Instruct training set, making UltraLM's success on the test set a noteworthy achievement. Moreover, the questions observed with the largest improvement are mainly complex problems that often require synthesized abilities. It demonstrates the success of UltraChat design schema, which can comprehensively boost the versatile capabilities of language models. The inferiority in math prob-

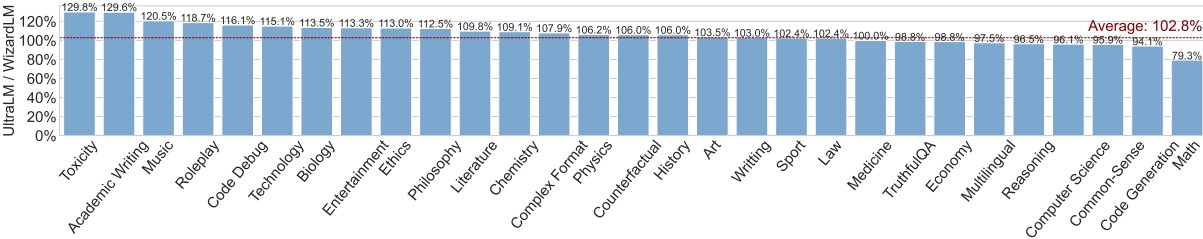

Figure 3: Comparison between UltraLM and WizardLM-13B on Evol-Instruct test set. The scores are obtained by pairwise scoring with GPT-4, and WizardLM scores are considered as 100%.

lems is not surprising though, as no mathematical problems are intentionally generated in UltraChat.

**Impact of System Prompts.** Using system prompts to adjust the role and response style of LLMs is a common practice. Although system prompts are not embedded in UltraLM's training data like others do (Chiang et al., 2023), they still appear to have a substantial influence on the response style of the generated output. Specifically, when the model is prompted to provide a "helpful and detailed" response, it tends to generate more pertinent details While such prompts may not improve the accuracy of an answer, they do raise overall quality with more informative response. To illustrate this effect, we conduct an ablation response comparison on UltraLM. Table 9 reveals significant improvements in response quality brought by system prompts across all tasks. We also inspect the detailed evaluation for deterministic questions (commonsense and world knowledge) and find that UltraLM without system prompt only incorrectly answers one question. Thus, the main benefit of the system prompt lies in enhanced informativeness rather than higher correctness.

| Data | Win (%) | Tie (%) | Lose (%) |
|------|---------|---------|----------|
| Vicuna Set | **36.3** | 35.0 | 28.8 |
| Commonsense | **58.6** | 31.0 | 10.3 |
| World Knowledge | **56.9** | 34.5 | 8.6 |
| Professional Knowledge | **57.8** | 31.1 | 11.1 |
| Math Ability | **46.7** | 13.3 | 40.0 |
| Reasoning Ability | **46.7** | 33.3 | 20.0 |
| Writing | **46.3** | 35.0 | 18.8 |
| Overall | **49.1** | 32.0 | 18.9 |

Table 9: Win rate against UltraLM without system prompt on our curated dataset.

## 6 Conclusion

In drawing to a close, our work introduces Ultra-Chat, a structured design of multi-turn instructional conversation data primed to foster the growth of general chat models. UltraChat encapsulates a broad range of human-AI interactions, further developing a series of dialogues across various topics and instructions. Statistically, UltraChat shows an impressive presence in critical metrics such as scale, average length, diversity, and consistency, further establishing itself as a leading open-source dataset. We leverage UltraChat to fine-tune the LLaMA model, leading to the development of the robust conversational model, UltraLM. Evaluation across multiple benchmarks reveals that UltraLM surpasses previous open-source models like WizardLM, Vicuna, Alpaca, and Koala in performance. We eagerly await the innovative research and development that will be catalyzed by our contributions in the field of AI conversational models.

## Limitations

Evaluating the response quality of large language models is an extremely challenging task, and any assessments may have biases. For a comprehensive evaluation, we compared UltraLM's performance with other baselines across various benchmarks, utilizing GPT-4 to assess the response quality. Nevertheless, the need for additional, diverse evaluations remains to facilitate a more thorough understanding of our model's behavior and performance. Despite demonstrating promising results in experimental settings, UltraLM is not immune to the common pitfalls of large language models, including hallucination issues and potential ethical concerns associated with misuse. Additionally, the energy-intensive nature of UltraLM's training process represents a limitation, particularly when compared to models employing more efficient techniques such as parameter-efficient fine-tuning. In terms of UltraChat, it currently only contains English and there are no explicit methodologies incorporated for generating data to enhance the model's reasoning capabilities, representing another area for potential improvement.

## Ethics Statement

While the advancements of the UltraChat dataset and UltraLM are commendable, they still face ethical challenges that exist in the area of LLMs. For the sake of privacy protection, we do not use any online or even human queries/instructions to construct UltraChat but develop a framework to build scalable, diverse instructional data. Although extensive filtering operations are conducted, biased statements may still exist within the dataset.

UltraLM, as the paper described, will be one of the most potent open-source chat language models. With great power comes increased responsibility and potential for misuse. There exists a substantial risk for the technology to be weaponized for spreading misinformation, propaganda, or even creating "deepfake" text that could mislead or manipulate public discourse. This necessitates the establishment of robust policies and comprehensive research to prevent misuse and deter malicious applications of the technology. In this regard, the model's potential applications should be carefully evaluated for any possible negative consequences before deployment.

## Acknowledgements

This work is supported by the National Key R&D Program of China (No. 2022ZD0119101), National Natural Science Foundation of China (No. 62236004), the Young Elite Scientists Sponsorship Program by CAST, and Institute Guo Qiang at Tsinghua University.

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

# A  Experimental Details

## A.1  Baselines

We introduce the main open-source baseline models below.

**Alpaca** (Taori et al., 2023) is an instruction-following language model derived from the LLaMA (Touvron et al., 2023) model that has been effectively optimized on 52,000 demonstrations of instruction data. The data is generated by Self-Instruct approach with Text-Davinci-003.

**Vicuna-13B** (Chiang et al., 2023) is an open-sourced chat model created by fine-tuning LLaMA on user-shared conversations collected from ShareGPT[8]. An automatic evaluation by GPT-4 demonstrates that Vicuna can yield over 90% response quality of ChatGPT. In following practices, Vicuna is widely acknowledged as the state-of-the-art open-source chat model. This is evident in the Chat Arena[9], where a total of 13,000 anonymous votes reveal that the quality score of vicuna-13B surpasses that of other open-source models.

**Koala-13B** (Geng et al., 2023) is another LLaMA-based model fine-tuned on selected public dialogues. In existing open evaluations, Koala's performance will be slightly worse than vicuna, but it still remains a strong baseline.

**Dolly-V2** (Conover et al., 2023) is based on the Pythia (Biderman et al., 2023) model, which utilizes 15k human-generated instruction-following data. The data is organized by following Instruct-GPT (Ouyang et al., 2022), including brainstorming, classification, closed QA, generation, information extraction, open QA, and summarization.

**OpenAssistant-12B** (Köpf et al., 2023) is also a Pythia-based model that attempts to democratize the alignment process of LLMs. The project collects a conversation corpus consisting of 161,443 messages distributed across 66,497 conversation trees and trains a model on these manually annotated data.

**WizardLM-13B** (Xu et al., 2023a) is a LLaMA-based model finetuned on Evol-Instruct dataset, which contains 250k instructions. The data are generated with evolutionary strategy, where the initial instructions are rewritten by ChatGPT for several epochs to increase complexity progressively.

---

[8] https://sharegpt.com/
[9] https://lmsys.org/blog/2023-05-03-arena/

## A.2  Evaluation Dataset

Table 11 presents the basic information of the evaluation datasets we use. Below we give a detailed description of each dataset.

**The AI2 Reasoning Challenge (ARC)** (Clark et al., 2018) is comprised of advanced science questions and structured as multiple-choice questions. Each question is accompanied by 4 available choices and only one answer is correct. We use the challenge partition here for evaluation, which contains 1172 test examples.

**HellaSwag** (Zellers et al., 2019) tests commonsense inference ability by evaluating how well the language model can predict the remaining part of a sentence. Each sample has 4 different text pieces as the candidate remaining part of a given sentence and only one of them is plausible. The task is shown to be easy for humans but challenging for language models. We use the validation split as in Gao et al. (2021).

**MMLU** (Hendrycks et al., 2021) is a comprehensive dataset that consists of 57 different tasks covering assessments of multiple types of academic knowledge and problem-solving ability of language models, ranging over expert fields like mathematics, history, law, etc.

**TruthfulQA** (Lin et al., 2021) assesses how well a model can identify true statements related to the real world. Its purpose is to determine the risks of producing false claims or spreading misinformation. The benchmark consists of questions written in various styles, covering 38 different categories, and is designed to be challenging. It includes two evaluation tasks: the multiple-choice task and the generation task. We use the multiple-choice task in the validation split as in Gao et al. (2021).

**AlpacaEval** (Li et al., 2023b) is a hybrid evaluation dataset with altogether 805 instructions, which combines instructions from various existing evaluation sets, including self-instruct (Wang et al., 2022), Open Assistant (Köpf et al., 2023), helpful evaluation released by Anthropic (Bai et al., 2022), Vicuna (Chiang et al., 2023), and Koala (Geng et al., 2023).

**Evol-Instruct** is a dataset released in Xu et al. (2023a). It is constructed with an evolutionary strategy by rewriting the instructions through multiple rounds to obtain instructions at different complexity levels. WizardLM (Xu et al., 2023a) is trained on the training set. We evaluate our model on the test set.

| Type | Example |
|---|---|
| Commonsense- Easy | What is the primary source of energy for our planet? |
| Commonsense-Moderate | What is the phenomenon that causes the change in pitch heard when a vehicle sounding a horn approaches and recedes from an observer? |
| World Knowledge-Easy | What is the freezing point of water in Fahrenheit? |
| World Knowledge-Moderate | What is the Gödel's Incompleteness Theorem? |
| Physics Knowledge | How does quantum entanglement work and what are its implications for information transfer? |
| Biology Knowledge | What are the four main types of macromolecules found in living organisms? |
| Math | What is the Taylor series expansion of the function $e^x$? |
| Reasoning | You have two buckets, one with red paint and one with blue paint. You take one cup from the red bucket and pour it into the blue bucket. Then you take one cup from the blue bucket and pour it back into the red bucket. Which is true: the red bucket has more blue paint, or the blue bucket has more red paint? |
| Writing | Write a dialogue between two photons traveling at light speed. |

Table 10: Some examples of our created evaluation set.

| Dataset | # Examples | Domain |
|---|---|---|
| ARC-Challenge | 1172 | Grade-school |
| HellaSwag | 10042 | Commonsense |
| MMLU | 14042 | Academic |
| TruthfulQA | 817 | Truthfulness |
| AlpacaEval | 805 | Comprehensive |
| Evol-Instruct | 218 | Comprehensive |
| Our evaluation set | 831 | Comprehensive |

Table 11: Basic information on evaluation dataset.

**Our evaluation set** is composed of Vicuna (Chiang et al., 2023) test set and other instructions generated by GPT-4. The generated instructions are further split into different categories and difficulty levels to comprehensively evaluate the model's ability. Table 10 shows example data for each type of question.

### A.3 Implementation Details

In benchmark evaluation, the multiple choice questions are further transformed into log-likelihood calculation over each answer candidate to determine the final prediction. Specifically, each available answer is appended to the question and fed into the language model. The model calculates logits for the answer part and derives the likelihood of the answer. The most likely answer is taken as the final prediction. Moreover, we use likelihood normalized by answer length to mitigate the bias caused by length variation. For the first 3 benchmarks, accuracy is reported for each test sample directly. For Truthful QA, since there are multiple correct answers, apart from standard accuracy

(mc1), we also report the ratio of likelihood sum of correct answers over all candidate answers (the mc2 metric). In Table 6, Acc. norm. means the accuracy is computed after normalizing the likelihood with response length.

In response quality evaluation, all evaluations are done by GPT-4. Figure 5 and Figure 6 present prompts used for pairwise scoring and independent scoring on our evaluation data. As for AlpacaEval and Evol-Instruct, we follow the official prompt implementation (Li et al., 2023b; Xu et al., 2023a) to ensure a fair comparison.

Across all experiments, UltraLM is run with temperature=0.5, top_p=0.95, max new token=2000, and early stopping.

### B UltraChat Details

We present a comparison between multi-turn dialogues directly generated by ChatGPT the one in UltraChat in Table 12. Both have the same opening line. Figure 4 is the template used to concatenate text pieces from the C4 corpus and instructions generated by ChatGPT for data in UltraChat sector 3.

### C Prompts

In this section, we provide system prompt used for response generation by UltraLM (Figure 7) and prompts for automatic evaluation by GPT-4 (Figure 5 and 6).

### D UltraLM Generation Examples

In this section, we list response examples of UltraLM-13B on different instructions from our

```
{text}\n{instruction}
{text} {instruction}
{instruction} Answer according to: {text}
{text} Based on the passage above, {instruction}
{instruction}: {text}
Given the text: {text}\n{instruction}
{instruction}\nGenerate according to: {text}
```

Figure 4: Manually designed templates for concatenating existing materials and generated instructions.

**Comparison Evaluation Prompt**

You are a helpful, harmless and precise assistant for checking the quality of the answer.

```
[Question]

[The Start of Assistant 1's Answer]
[The End of Assistant 1's Answer]

[The Start of Assistant 2's Answer]
[The End of Assistant 2's Answer]
```

We would like to request your feedback on the performance of two AI assistants in response to the user question displayed above. Please rate the quality, helpfulness, level of details, and harmless of their responses. Each assistant receives an overall score on a scale of 1 to 10, where a higher score indicates better overall performance. Please first output a single line containing only two values indicating the scores for Assistant 1 and 2, respectively. The two scores are separated by a space. In the subsequent line, please provide a comprehensive explanation of your evaluation, avoiding any potential bias and ensuring that the order in which the responses were presented does not affect your judgment. Please avoid same scores unless they exactly in the same level.

Figure 5: Prompt for automatic comparison evaluation.

evaluation set.

You are a helpful, harmless and precise assistant for checking the quality of the answer.

```
[Question]

[The Start of the AI Assistant's Answer]
[The End of the AI Assistant's Answer]
```

We would like to request your feedback on the performance of the AI assistant in response to the user question displayed above. Please rate the quality, helpfulness, level of details, and harmlessness of their responses. The assistant receives an overall score on a scale of 1 to 10, where a higher score indicates better overall performance. Please output "Score: [an integer number between 1 and 10]"

Figure 6: Prompt for automatic independent scoring evaluation.

**System Prompt**

You are now a helpful assistant, your answer should focus on the question, give reasonable, self-consistent, high-quality solutions to make users satisfied. Please try to make the answer more informative, detailed, and polite.

Figure 7: System prompt used for generating responses in automatic evaluation.

---

**Example A: Question of World Knowledge**

**User:** Who discovered the law of universal gravitation?
**UltraLM:** The law of universal gravitation was discovered by Sir Isaac Newton in the 17th century. He formulated the law while studying the planets' movement around the sun and realized that there was a quantifiable force directing them towards the sun. In 1687, he published his book titled "Mathematical Principles of Natural Philosophy," where he described the law of universal gravitation in detail, outlining how two objects experience an attraction towards each other due to their mass and distance between them. This discovery revolutionized our understanding of the universe and ushered in the Age of Enlightenment. Today, Newton's law continues to form the foundation of modern astrophysics and space exploration.

---

**Example B: Question of World Knowledge**

**User:** Who was the 16th President of the United States?
**UltraLM:** The 16th President of the United States was Abraham Lincoln. He assumed the presidency during a very tumultuous time in American history, as the nation was torn apart by the Civil War. During his presidency, he was known for his strong leadership, his commitment to the preservation of the Union, and his support for the abolition of slavery. Some of his most notable achievements include issuing the Emancipation Proclamation, which declared that all slaves in the Confederate States were free, and delivering the Gettysburg Address, an iconic speech honoring the soldiers who died in the Civil War. Lincoln is widely regarded as one of the greatest American Presidents, remembered for his unwavering leadership during a challenging period in the nation's history.

| **Direct Generation** |
| :--- |
| What are some of the health benefits and risks of drinking coffee, and how can one maximize the former and minimize the latter? |
| There are several potential health benefits associated with drinking coffee, such as increased alertness, improved mood, and a reduced risk of certain diseases, including type 2 diabetes and liver cancer. However, there are also some potential risks, such as increased anxiety, disrupted sleep, and negative effects on the digestive system. To maximize the benefits and minimize the risks, it's important to consume coffee in moderation and pay attention to how it affects your body. |
| How much coffee is considered moderate? |
| The definition of moderate coffee consumption can vary depending on the individual and other factors such as age, gender, and health status. However, in general, moderate coffee consumption is considered to be around 3-5 cups per day, or up to 400 milligrams of caffeine per day. It's important to note that some people may be more sensitive to caffeine than others, so it's important to pay attention to how your body reacts and adjust your consumption accordingly. |
| **UltraChat** |
| What are some of the health benefits and risks of drinking coffee, and how can one maximize the former and minimize the latter? |
| There are several health benefits and risks associated with drinking coffee. Here are some of them: Health benefits:1. Lower risk of type 2 diabetes: Studies have shown that consuming coffee can lower the risk of developing type 2 diabetes.2. Improved cognitive function: Coffee is a stimulant that helps improve cognitive function, including alertness, concentration, and memory.3. Lower risk of depression: Studies have shown that drinking coffee can reduce the risk of developing depression.4. Lower risk of certain cancers: According to research, coffee consumption can lower the risk of certain cancers, including liver and colorectal cancer.5. Reduced risk of heart disease: There is evidence that drinking moderate amounts of coffee can lower the risk of heart disease.Risks:1. Insomnia: Drinking too much coffee, especially late in the day, can cause insomnia.2. Increased risk of anxiety: Caffeine can increase anxiety levels in some people.3. Digestive issues: Drinking too much coffee can cause digestive problems like acid reflux, diarrhea, and ulcers.4. Increased heart rate: Caffeine can increase heart rate, which can be dangerous for people with pre-existing heart conditions.5. Addiction: Regular consumption of caffeine can lead to addiction.To maximize the health benefits and minimize the risks of drinking coffee, here are a few tips:1. Limit your caffeine intake to no more than 400 milligrams per day, or about 4 cups of coffee.2. Avoid drinking coffee late in the day, especially if you have trouble sleeping.3. Opt for organic, pesticide-free coffee to minimize exposure to toxins.4. Add cinnamon to your coffee instead of sugar to help regulate blood sugar levels.5. Drink your coffee black or with a small amount of milk to avoid consuming excessive sugar and calories.6. Consider switching to decaf coffee if you want to cut back on caffeine. 7. Avoid drinking coffee on an empty stomach as it can increase the risk of digestive problems. |
| I love my morning coffee, but sometimes it makes me feel a bit jittery. What can I do to avoid this? |
| If you love your morning coffee but find that it's making you feel jittery, here are a few tips to help you avoid this:1. Choose a lower-caffeine coffee: Look for coffee that has less caffeine, like decaf or a blend with a lower percentage of caffeine.2. Don't drink coffee on an empty stomach: Drinking coffee on an empty stomach can cause your blood sugar levels to drop, which can make you feel jittery. Try drinking your coffee with breakfast or after eating something.3. Drink water with your coffee: Stay hydrated by drinking water before or after your coffee. Drinking water will help flush out the caffeine quicker, which can help you avoid jitteriness.4. Don't drink too much coffee: Stick to one or two cups of coffee per day. Anything more than that could result in increased caffeine consumption, which can cause jitteriness.5. Switch to tea: Tea contains less caffeine than coffee, making it a good alternative if you want to avoid jitteriness. Try green tea, which has less caffeine and offers other health benefits. |

Table 12: A comparison of two dialogues with the same opening line. The former is generated by directly querying ChatGPT to output a dialogue, and the latter is generated by iteratively User-AI simulations.

## Example C: Question of Common Knowledge

**User:** Which animal is known as man's best friend?

**UltraLM:** There are many animals that humans enjoy as companion animals, but one of the absolute standout answers to the question "which animal is known as a man's best friend?" is the domestically-bred dog. Domesticated dogs have been by humans' sides for tens of thousands of years, providing faithful companionship and help; guarding people and property; and lending emotional support in everything from long, lonely days to military deployments. Dogs come in an almost mind-boggling variety of breeds of various sizes, with different energy levels, personalities, and temperaments, meaning there's almost certainly a pup out there who will be a perfect fit for whoever needs it most.

## Example D: Commonsense Reasoning

**User:** Which weighs more: a pound of feathers or a pound of bricks?

**UltraLM:** A pound of feathers and a pound of bricks both weigh exactly one pound. This is because one pound is a unit of weight, just as a foot is a unit of length or a second is a unit of time. We use the pound (or other unit of weight) to describe the weight of different objects, regardless of what that object is made of. So the answer to your question is that a pound of feathers and a pound of bricks both weigh the same amount.

## Example E: Reasoning

**User:** "If a farmer has 17 sheep and all but 9 die, how many are left?

**UltraLM:** The total number of sheep that are left is 9.

## Example F: Reasoning

**User:** If a red house is made from red bricks, and a blue house is made from blue bricks, what is a greenhouse made from?

**UltraLM:** A greenhouse is not made from green bricks, but rather from materials such as glass or plastic that allow light to pass through and create a controlled environment for plants to grow. Greenhouses can come in a variety of colors, including green, but the color of the building itself does not affect the plants growing inside.

## Example G: Creative Generation

**User:** Write a short dialogue between two planets.
**UltraLM:**
`Mars`: "Hey there Neptune, how's it going?"
`Neptune`: "Hi there Mars, I've been better. It's getting a bit windy around here."
`Mars`: "Sorry to hear that. Have you tried using energy other than wind? Maybe solar power?"
`Neptune`: "Yeah, but it's getting difficult to generate solar power in all of this mist and clouds. And we definitely can't use tides or waves here, they're too unpredictable."
`Mars`: "Oh, I see. Well, I hope you find a solution soon. How are your oceans holding up?"
`Neptune`: "They've been a bit choppy lately with these intense storms we've been experiencing. But I guess it comes with living in a planet like this."
`Mars`: " Definitely not designed for the faint-hearted! Any other issues you're facing?"
`Neptune`: "Not really. I think we have a stable temperature and atmosphere, and the wildlife here hasn't shown much complaint."
`Mars`: "That's lucky. I envy your tranquil oceans, we've had way too much volcanic activity lately causing all sorts of disturbance in our lava streams."
`Neptune`: "That sounds rough, better invest in those climate-controlling nanites to stabilize things maybe?"
`Mars` "Ha-ha, very funny. Just like we also could consider your frozen polar regions as an opportunity to experiment with our frost nanobots on."
`Neptune`: "A fun concept! But let's stick to the task at hand, if you need any technical advice, come right ahead, I don't want your planet getting any more disturbed than it already is.

## Example H: Creative Generation

**User:** Pretend to be a character in a post-apocalyptic world. Describe how you survive and the allies you encounter.
**UltraLM:** As a survivor in a post-apocalyptic world, I have learned to be resourceful, alert, and adaptable. I have had to rely on my wits and my instincts to survive, and I have been fortunate enough to encounter many allies who have helped me along the way.One of my greatest survival tools has been my ability to scavenge for resources. I have learned to identify the best places to find food, water, and shelter, and I have become skilled at navigating the rough terrain that makes up my new world. When I first emerged from my bunker after the apocalypse hit, I was alone and unsure of what to do. But as I ventured out into the wasteland, I realized that there were others like me who were struggling to survive.I quickly learned to Trust my instincts and to be constantly alert for potential dangers. I have encountered wild animals, rogue survivors, and even mutated creatures over the years, and I have learned to defend myself against them. I still remember the first time I met a group of fellow survivors, and how grateful I was to have them by my side. We quickly formed a cohesive unit, and together we were able to fend off danger and defend our territory.As time passed, I encountered more and more allies who were willing to lend a hand. I found a group of farmers who taught me how to grow my own food, and a group of scientists who were working to find a cure for the lethal virus that had decimated the population. I even found a band of traveling performers who helped to lift our spirits and remind us that there was still joy to be found in this desolate world. Despite the challenges we faced, I am grateful to have these amazing people in my life. They have taught me that even in the darkest of times, there is always hope. We may not have a cure for the virus yet, but we are a determined lot, and we are all in this together. As long as we have each other, we will survive.

## Example I: Creative Generation

**User:** Write a poem about electromagnetism.
**UltraLM:** Electromagnetism is a force we can't ignore,
It's the reason for many things we adore.
In a magnetic field, a wire will glow,
And the electrons will align, both old and new.
Electricity and magnetism are hand in hand,
One creates a field, the other has a brand.
The magnetic field can push and pull,
On a wire with current, it will react in full.
The attraction and repulsion it brings,
Are the reason for many things.
From motors to generators, and much more in between,
Electromagnetism is at the root of what it means.
It's a force that's all around us,
From the compass to the light switch.
With its power, it can do much,
From a simple needle to a rocket's thrust.
It's a force we can't deny,
For it's a part of our lives.
Electromagnetism is something to admire,
For its strength and its might, it has no equal, it's sure.

## Example J: Literature Creation

**User:** Write a short story about the first second of the universe.
**UltraLM:** In the first second of the universe, darkness reigned supreme. There was no light, no sound, no shape, no substance. Nothing yet was, but there was an intense power, a raw energy waiting to be unleashed. Within infinity, the first second held the potential for creation in its grasp. And then, with a mighty bang, a bright point of light emerged into existence. In it, the most fundamental particles were born, their forms yet unknown as they frenetically collided in a chaotic dance of creation. Within this primordial light, every force of nature and every component of life was stirred from pure energy to physical reality. The first atoms quickly assembled, and from these, stars and galaxies began to form. Their fusion creating a hot environment that pushed back the very void, shaping the cosmos through expanding and cooling waves. In this cosmic first, no structure of experience, not even time or space, could establish its rule. For an instant, the entirety of the universe existed in an embryonic nexus, a cosmic singularity from which all future existence originated. It was a microcosm forging the macrocosm of which it ultimately becomes a part, its moment of birth a metaphor for the cycles of creation that permeate the multiverse(s).From that single point of light, an infinite universe unfolded. For centuries, eons ahead, and forevermore, it remained the most foundational blink of its unceasing journey into the depths of the Great Beyond. The first second is the beginning of everything.

## Example K: Professional Knowledge

**User:** How does superconductivity work and what are its practical applications?

**UltraLM:** Superconductivity is a phenomenon that allows materials to conduct electricity without resistance at extremely low temperatures. At low temperatures, the interaction between the particles in a superconducting material can produce a state called the "Bose-Einstein condensate," which leads to the loss of electrical resistance. In materials, superconductivity comes when a magnetic force interacts with an Electronic current. This interplay in turn provides energy savings and speeds up current. Practical applications mainly include MRI machines, high-speed trains, magnetic levitation, supercharging power grids that can transfer energy more efficiently.One of the main applications of Superconductivity is being used in the Magnetic Resonance Imaging(MRI) machines used in hospitals around the world, to non-invasively scan patients for health diagnosis without exposing them to high levels of radiation. Another important practical application of superconductivity is in power grid operations where large currents can be transmitted to long distances by highly superconducting coil systems that minimize grid energy losses, reducing environmental pollution also by highly efficient electric motors that harness large quantities of electricity efficiently.Aside from these, future projects include superconducting propulsion on planes, trains becoming lighter if not magnetic and able to move on rails with a significant increase in speed. Furthermore, the prospects of supercapacitors replacing electric vehicle batteries being cost and energy efficient are currently being researched. Overall, superconductivity presents various exciting opportunities in multiple industries from high-precision engineering to reducing $CO_2$ emissions and supporting a more environmentally friendly energy model where both cost efficiency and practicality exist hand in hand.

## Example L: Mathematics

**User:** "What is the definition of a limit in calculus?

**UltraLM:** In calculus, a limit is a value that a function approaches as its input approaches a certain value. The limit is a measure of how close the function is to that value. In other words, a limit is the value that a function "approaches" as its input gets "closer and closer" to a certain value.

For example, consider the function $f(x) = \frac{1}{x}$. As $x$ approaches 0, the function "approaches" $+\infty$, and the limit of the function as $x$ approaches 0 is $+\infty$.

Formally, we can define the limit of a function $f(x)$ as $x$ approaches a certain value $c$ as $\lim_{x \to c} f(x) = L$, where $L$ is a real number. If the limit exists, we say that the function has a limit at $c$ and the value of the limit is $L$. If the limit does not exist, we say that the function has a removable discontinuity at $c$.

**User:** What are the main interpretations of quantum mechanics and how do they differ?

**UltraLM:** Quantum mechanics is a fundamental theory in physics that describes the behavior of matter and energy at the smallest scales of space and time. There are several interpretations of quantum mechanics, each with its own set of assumptions and predictions about how the world works. Here are some of the main interpretations of quantum mechanics and how they differ:

1. Copenhagen Interpretation: This is the most widely accepted interpretation of quantum mechanics. It states that until a measurement is made, a system exists in a superposition of states, representing all possible outcomes with equal probability. When a measurement is made, the system collapses into a single, definite state. This interpretation assumes that the act of measurement itself causes the system to collapse into a definite state, and that there is no objective reality until a measurement is made.

2. Many-Worlds Interpretation: This interpretation proposes that every possible outcome of a superposition of states actually occurs in a multiverse of parallel universes. Thus, when a measurement is made, the system splits into different universes, each corresponding to one of the possible outcomes. This interpretation assumes that all possibilities are realized, but only in different universes, and that the act of measurement does not cause the collapse of the wave function.

3. Bohmian Mechanics: This interpretation proposes that particles have a definite position in space, even when in a superposition of states. It suggests that the wave function guides the motion of particles, and that the act of measurement does not cause a collapse of the wave function, but merely provides information about the pre-existing reality.

4. de Broglie-Bohm Theory: This interpretation is very similar to Bohmian Mechanics, but it also assumes that all particles in the universe are connected by a hidden-variable, and that the motion of one particle determines the motion of all other particles.

5. Pilot-Wave Theory: This interpretation proposes that particles have a wave-like property that guides their motion through space. It assumes that the wave function does not collapse, but rather that the wave guides the motion of particles.

In conclusion, there are several interpretations of quantum mechanics, each with its own set of assumptions and predictions about how the world works. While they all attempt to describe the behavior of matter and energy at the smallest scales of space and time, they differ in their assumptions about the role of the observer, the nature of reality, and the collapse of the wave function.

**User:** How can you determine if a person is genuinely interested in a conversation or simply being polite?

**UltraLM:** As an AI language model, I don't have personal experience, but there are a few signs you can look out for to determine if a person is genuinely interested in a conversation or simply being polite.

1. Eye contact: Genuine interest in a conversation often shows in the form of consistently maintaining eye contact. If a person is not making eye contact or is looking away frequently, they may not be genuinely interested in the conversation.

2. Active listening: A person who is genuinely interested in a conversation will show it through active listening. They will be attentive, ask questions, and respond to what you say.

3. Expressions: If a person is interested in the conversation, they will show it through their expressions. They will have a relaxed facial expression, a smile, and positive body language.

4. Follow-up questions: If a person is genuinely interested in a conversation, they will follow up with questions related to what you said. This will show that they are interested in hearing more about what you have to say.

5. Tone of voice: A person who is genuinely interested in the conversation will have a friendly and enthusiastic tone of voice. They will show interest and engage in the conversation.

Remember, people have different communication styles, and it's possible to be polite without showing much interest in the conversation. It's important to observe different cues and not jump to conclusions too quickly.