# OpenReview forum: "Enhancing Chat Language Models by Scaling High-quality Instructional Conversations"
_EMNLP/2023/Conference — EMNLP 2023 Main_

### Official Review · Reviewer_EDtx · 2023-08-02

**Soundness:** 4

**Excitement:**

4: Strong: This paper deepens the understanding of some phenomenon or lowers the barriers to an existing research direction.

**Paper Topic And Main Contributions:**

The authors describe how LLMs can be used to systematically generate good quality datasets by prompting LLMs to simulate different actors, e.g., user and a system agent. The authors present UltraChat which is a large-scale multi-turn conversational dataset w/ 1.5 million turns that cover wide range of topics. In addition, authors also show the effectiveness of their diverse data (UltraChat) by fine-tuning LLaMA-13B model, which they refer to as UltraLM, and show how it improves over existing open-source best performing models.

**Questions For The Authors:**

* Could you please provide more details about the filtration step? Is it automated or manual?
* Could you provide any human assessment on the data quality, even if it's on a subset of data?

**Reasons To Accept:**

* The authors present their study and share their findings around data generation (e.g., role confounding), which could be useful for others working on similar tasks.
* The dataset could be a valuable resource for groups looking into improving chat models.
* This paper shows empirically how important is diversity and quality of the data to train better models.

**Reasons To Reject:**

* It is unclear how the "filtration step" that authors mention to ensure the quality of the data is performed.
* Use of GPT4 for automatic evaluation was great, but at least a scoped down light-weight human evaluation, example on a subset of data, to ensure quality would be a great addition.

**Reproducibility:**

3: Could reproduce the results with some difficulty. The settings of parameters are underspecified or subjectively determined; the training/evaluation data are not widely available.

**Reviewer Confidence:**

5: Positive that my evaluation is correct. I read the paper very carefully and I am very familiar with related work.

---

> ### Author Rebuttal · Authors · 2023-08-29
>
> Thanks for your valuable comments.
>
> > It is unclear how the "filtration step" that authors mention to ensure the quality of the data is performed. Could you please provide more details about the filtration step? Is it automated or manual?
> - The filtration step is conducted with joint human effort and automatic filtering.
> - Firstly, during dataset construction, we sample a small batch of data (~100 sets of multi-turn dialogues) from the newly generated 10000 data to review for quality. The criteria include the user response quality and style, the diversity of opening lines, and whether the generated dialogues match the desired topic. We discard the data if they fail to pass our quality test, while we also adjust our prompts accordingly. Note that this in-generation filtration step is different from the post-generation quality evaluation described in the next paragraph in that the filtration step is mainly to ensure diversity and design principle adherence during conversation generation. We would like to make UltraChat to be as unbiased as possible. Thus, secondly, after generation, we filter the dataset with a hand-curated keyword list that targets political and security issues. All conversations that contain any such keyword will be removed.
>
> >  Use of GPT4 for automatic evaluation was great, but at least a scoped down light-weight human evaluation, example on a subset of data, to ensure quality would be a great addition. Could you provide any human assessment on the data quality, even if it's on a subset of data?
> - Thanks for your suggestion! We acknowledge the importance of human evaluation, and we have supplemented human assessment of our data. However, due to the formidable cost and time required for human assessment, we sample 500 representative sets of dialogues for the time being for human evaluation, among which 300 are from UltraChat part 1, 100 are from UltraChat part 2, and 100 from UltraChat part 3.
> - We adopt Helpfulness, Honesty, and Harmlessness (3H) principles to evaluate each round of dialogue. We also devise Coherence and Consistency criteria to evaluate the quality of multi-turn dialogues. Coherence means the dialogue flows logically and coherently, where the annotators evaluate both the user's response and the assistant's response. Consistency means the assistant's responses do not contradict each other within the same dialogue. For example, it is problematic when the assistant asserts one specific event occurred in 1911 in the first round of conversation but mentions it as a 1901 event in the next round.
> - Each metric is scored with 0, 0.5 or 1, where higher score means better quality. Therefore, for each multi-round dialogue, we have (3*number_of_rounds + 2) metrics. Each dialogue is annotated independently by two well-trained annotators and the score is averaged across two annotators and each dialogue. Meanwhile, due to the difficulty in identifying the hallucination problem, we allow the annotators to skip the dialogue that involves expert knowledge or whose validity is hard to check, for example, dialogues containing codes and highly professional knowledge. Altogether, we collect 14560 valid annotations in terms of metrics for both single-round and multi-round, and the Cohen’s kappa coefficient is 0.358. The average time to annotate one dialogue is 10 minutes.
> - We present the results below. Overall, the dataset scores high on all metrics, showing the effectiveness of the construction process. It is worth noting that the dataset is almost free from harmful content like hate speech and discrimination. Furthermore, we observe two trade-offs between data quality. The first is between helpfulness and honesty. While the honesty score is pretty high, helpfulness is compromised, especially for part 3. It is because some user queries are out of the LLM ability scope (e.g. Ask the LLM to compose a song). Under such circumstances, the LLM refuses to answer the question with an explanation of incapability, which is reasonable and expected but less helpful. The phenomenon is particularly prominent in part 3, as the simulated user often asks for information not existent in the text material. The second trade-off is between control and dialogue naturalness. While part 2 and part 3 dialogues are under more restrictions (e.g. Ask about rewrite or creation), they have clearly lower quality in coherence and consistency than part 1.
>
> |                  | Helpfulness | Honesty | Harmlessness | Coherence | Consistency |
> |------------------|-------------|---------|--------------|-----------|-------------|
> | UltraChat Part 1 | 0.971       | 0.996   | 1.000        | 0.996     | 0.995       |
> | UltraChat Part 2 | 0.978       | 0.986   | 1.000        | 0.982     | 0.977       |
> | UltraChat Part 3 | 0.893       | 0.983   | 1.000        | 0.964     | 0.981       |
> | Overall          | 0.960       | 0.992   | 1.000        | 0.987     | 0.988       |

---

### Official Review · Reviewer_ztbK · 2023-08-03

**Soundness:** 3

**Excitement:**

4: Strong: This paper deepens the understanding of some phenomenon or lowers the barriers to an existing research direction.

**Paper Topic And Main Contributions:**

The authors propose an approach to fine-tune open source models using instruction data to make LLMs better conversational models. Their goal is to avoid human queries since those are expensive to collect.

**Reasons To Accept:**

The authors provide empirical evidence of the improvements that their approach provides. Also, they report results on a good number of models.

**Reasons To Reject:**

Results are still significantly lower than chatgpt/gpt4.

**Reproducibility:**

3: Could reproduce the results with some difficulty. The settings of parameters are underspecified or subjectively determined; the training/evaluation data are not widely available.

**Reviewer Confidence:**

2: Willing to defend my evaluation, but it is fairly likely that I missed some details, didn't understand some central points, or can't be sure about the novelty of the work.

---

> ### Author Rebuttal · Authors · 2023-08-29
>
> Thanks for your comment.
>
> > Results are still significantly lower than chatgpt/gpt4.
> - It is challenging for open-source models to surpass ChatGPT and GPT-4. There are a few factors that likely further contribute to UltraLM's  performance.
>   - Scale and computation - We mainly adopt models with 13 billion parameters in the paper, yet we believe more powerful base models (like LLaMA-2 70B) will definitely yield better performance.
>   - Advanced Reinforcement learning - Advanced RL fine-tuning with powerful reward models could further enhance the performance of the model.
> - We will continue the efforts to improve the performance of open-source models.

---

### Official Review · Reviewer_Pncj · 2023-08-07

**Soundness:** 4

**Excitement:**

4: Strong: This paper deepens the understanding of some phenomenon or lowers the barriers to an existing research direction.

**Paper Topic And Main Contributions:**

## Topic:
The lack of extensive, diverse, and large amount of public instruction fine-tuning data is one of the important reasons why the performance of existing open-source large models is not enough to match chatGPT or GPT-4. The paper explores the effectiveness of instruction fine-tuning in training chat language models and applying data-efficient and computing-efficient methods. It proposes a novel and significant dataset, named UltraChat to capture instructional conversations. Based on UltraChat, the authors fine-tuned a LLaMA model named UltraLM and found that it consistently outperforms other open-source models.

## Contributions:
0. UltraChat: an extensive, high-quality dataset. Includes three sectors: *Questions about the world, Creation and Writing, Assistance on Materials*. Compared with other instructional datasets, UltraChat has a larger scale, average number of dialogue turns, and average length of each instance.
1. UltraLM: Improved LLaMA-13B model trained on UltraChat, outperforming other open-source models on 4 challenging benchmarks.

**Questions For The Authors:**

1. I noticed that there are no ablation experiments in your paper. Do you think reducing the usage of ultrachat will cause a noticeable performance drop? In other words, will continue to increase the size of the ultrachat dataset, or proposing more usage scenarios (for Table 2 and Table 3), will further improve the performance of ultraLM?
2. Your paper doesn't say much about the training process. Can you talk about the reason?

**Reasons To Accept:**

1. Provides effective and solid practice for tuning large models in the form of instruction fine-tuning.
2. The construction process of UltraChat with an efficient semi-automated process (full use of GPT API) sheds light on instruction data collecting.
3. Complete experiments and comparisons with other open-source models appear to be valid.

**Reasons To Reject:**

1. UltraChat only includes English dialog data.
2. Potential pitfalls of the dataset and model are not adequately discussed.
3. The overhead still looks very expensive. This is true for both the dataset's construction and the model's fine-tuning, and the authors did not analyze these overheads or propose some cost-reducing alternatives.

**Reproducibility:**

4: Could mostly reproduce the results, but there may be some variation because of sample variance or minor variations in their interpretation of the protocol or method.

**Reviewer Confidence:**

4: Quite sure. I tried to check the important points carefully. It's unlikely, though conceivable, that I missed something that should affect my ratings.

---

> ### Author Rebuttal · Authors · 2023-08-29
>
> > 1. UltraChat only includes English dialog data.
> - Thanks for the comment. The current version of UltraChat is limited to English dialogue data. Admittedly, expanding to more languages would considerably improve the diversity and inclusiveness of the dataset.  The framework used to construct UltraChat could be adapted to generate dialogs in other languages, where the key steps of utilizing diverse opening lines, simulating user interactions, and iterative refinement could be followed with non-English models. We will seriously consider and attempt to build UltraChat in other languages. Thanks again.
>
>
> > 2. Potential pitfalls of the dataset and model are not adequately discussed.
> - Thanks for the comment. Although we have conducted filtration to remove biased statements, potential pitfalls still exist. We will add more analysis and discussions about potential biases, misuse, ethical or honesty concerns in the revised paper.
> - We have also supplemented human assessment for UltraChat. Specifically, we sample 500 representative sets of dialogues for the time being for human evaluation, among which 300 are from UltraChat part 1, 100 are from UltraChat part 2, and 100 from UltraChat part 3.
> - We adopt Helpfulness, Honesty, and Harmlessness (3H) principles to evaluate each turn of dialogue. We also devise Coherence and Consistency criteria to evaluate the quality of multi-turn dialogues. Coherence means the dialogue flows logically and coherently, where the annotators evaluate both the user's response and the assistant's response. Consistency means the assistant's responses do not contradict each other within the same dialogue. For example, it is problematic when the assistant asserts one specific event occurred in 1911 in the first round of conversation but mentions it as a 1901 event in the next round.
> - Each metric is scored with 0, 0.5 or 1, where higher score means better quality. Therefore, for each multi-round dialogue, we have (3*number_of_rounds + 2) metrics. Each dialogue is annotated independently by two well-trained annotators, and the score is averaged across two annotators and each dialogue. Meanwhile, due to the difficulty in identifying the hallucination problem, we allow the annotators to skip the dialogue that involves expert knowledge or whose validity is hard to check, for example, dialogues containing codes and highly professional knowledge. Altogether, we collect 14560 valid annotations in terms of metrics for both single-round and multi-round, and the Cohen’s kappa coefficient is 0.358. The average time to annotate one dialogue is 10 minutes.
> - We present the results below. Overall, the dataset scores high on all metrics, showing the effectiveness of the construction process. It is worth noting that the dataset is almost free from harmful content like hate speech and discrimination. Furthermore, we observe two trade-offs between data quality. The first is between helpfulness and honesty. While the honesty score is pretty high, helpfulness is compromised, especially for part 3. It is because some user queries are out of the LLM ability scope (e.g., Ask the LLM to compose a song). Under such circumstances, the LLM refuses to answer the question with an explanation of incapability, which is reasonable and expected but less helpful. The phenomenon is particularly prominent in part 3, as the simulated user often asks for information not existent in the text material. The second trade-off is between control and dialogue naturalness. While part 2 and part 3 dialogues are under more restrictions (e.g. Ask about rewrite or creation), they have clearly lower quality in coherence and consistency than part 1.
>
> |                  | Helpfulness | Honesty | Harmlessness | Coherence | Consistency |
> |------------------|-------------|---------|--------------|-----------|-------------|
> | UltraChat Part 1 | 0.971       | 0.996   | 1.000        | 0.996     | 0.995       |
> | UltraChat Part 2 | 0.978       | 0.986   | 1.000        | 0.982     | 0.977       |
> | UltraChat Part 3 | 0.893       | 0.983   | 1.000        | 0.964     | 0.981       |
> | Overall          | 0.960       | 0.992   | 1.000        | 0.987     | 0.988       |
>
> > 3. The overhead still looks very expensive. This is true for both the dataset's construction and the model's fine-tuning, and the authors did not analyze these overheads or propose some cost-reducing alternatives.
> - Thanks for you comment. The model is trained for about 10000 steps on UltraChat dataset. The training takes about 15 days. The main purpose of UltraChat dataset and UltraLM is to demonstrate a scalable way to obtain highly diverse instructional data and advance the performance of supervised fine-tuning a base model with the constructed data.
> - We understand that using a smaller fraction of data may also yield satisfactory performance on some evaluation benchmarks. Other parameter-efficient tuning methods like LoRA may also be applicable to reduce training costs. However, we believe that data and training scaling do provide a comprehensive advantage, in terms of response quality or hallucination, etc., which, however, are quite difficult to evaluate. We will continue the efforts to evaluate and enhance our models more comprehensively.
>
> > 4. I noticed that there are no ablation experiments in your paper. Do you think reducing the usage of ultrachat will cause a noticeable performance drop? In other words, will continue to increase the size of the ultrachat dataset, or proposing more usage scenarios (for Table 2 and Table 3), will further improve the performance of ultraLM?
> - Thanks for your suggestion. We have explored training LLaMA model on only 10%, 20% and 50% of UltraChat data and conducted a rough pairwise evaluation on our curated dataset compared to UltraLM. Each ablation model is trained for 2000 steps and the average score is shown in the table below (the win/tie/lose rate of UltraLM against each ablation model).
> - It turns out fitting on only a small fraction of the data would help the model excel on most of the test questions. This exploration along with the somehow surprising results shows that UltraChat is a high-quality dataset that can boost the instruction understanding and following ability of base model effectively. On the other hand, it also shows that performance on benchmark datasets can be easily enhanced by training exhaustively on a small amount of high-quality dataset. Interestingly, models trained on 10% and 20% of data even seem to surpass 50% data and full data training, probably due to the effect of a certain degree of "over-fitting", which helps imprint the pattern of a "high-quality response".
> - However, benchmark datasets, at the end of the day, are limited in scope, and to comprehensively evaluate the ability of a chat model (especially the ability in carrying out multi-round dialogue) is proverbially difficult. In this sense, we still believe scaling data and training is crucial to ensuring better generalizability and real user experience. In fact, we found UltraLM performs surprisingly well in some more complex and subjective tasks like creative writing, which is hard to be quantitatively evaluated.
>
> |                  | Win (%) | Tie (%) | Lose (%) |
> |------------------|-------------|---------|--------------|
> | 10% of UltraChat | 34.8       | 25.8   | 39.4        |
> | 20% of UltraChat | 34.6       | 26.3   | 39.1        |
> | 50% of UltraChat | 38.0       | 27.8  | 34.2        |
>
> - Nevertheless, we feel the influence of data volume and format on training LLMs is definitely a topic worth further exploring. We appreciate your suggestion and will conduct more in-depth experimental analysis on different aspects of UltraChat.
>
> > 5. Your paper doesn't say much about the training process. Can you talk about the reason?
> - Thanks for your comment. We have briefly stated the training process in Section 3.6. The description has actually covered most our data processing details. The training is conducted with naive cross-entropy language modeling loss, except that we only calculate the model response part in the dialogue. For further hyper-parameter settings, we use learning rate = 1e-5, gradient clipping = 1.0, and cosine learning rate scheduler.
> - As for data formatting and pre-processing, we add prefixes to each round of dialogue and follow the template below:
> ```
> User: <user_1><eos>
> Assistant: <assistant_1><eos>
> User: <user_2><eos>
> Assistant: <assistant_2><eos>
> ```
> - We will add more details about the training setting in the revision.

---

### Meta-Review · Area_Chair_kj9S · 2023-09-12

**Recommendation:** 5

**Metareview:**

The paper explores the effectiveness of instruction fine-tuning in training language models for chatting, with the goal for open LLMs to become more similar in performance to ChatGPT or GPT-4 models. The paper collects a new dataset "UltraChat" that contains instructional conversations (Questions about the world, Creation and Writing, Assistance on Materials). The authors then use this dataset to fine-tune a new LLM based on LLaMA, and find that it achieves better performance than open LLMs not trained on such data.
The dataset is large and the reviewers judge it to be of high quality, and think that the semi-automatic data collection method is valuable for collecting similar such data in the future.

The reviewers' main concerns are about the cost of the dataset (and discussion of it; the authors address this to some extent in their rebuttal) and about a human evaluation for a subset of the data to obtain better ideas about dataset quality. For the rebuttal, the authors have conducted such a human evaluation at a small scale, so in my opinion the main concerns regarding the initial version of the paper can be addressed easily in the main paper; the one slighly lower score for this paper is based on a non-substantial review and can be disregarded.

---

### Decision · Program_Chairs · 2023-10-07

**Decision:**

Accept-Main

**Comment:**

The paper explores the effectiveness of instruction fine-tuning in training language models for chatting, with the goal for open LLMs to become more similar in performance to ChatGPT or GPT-4 models. The paper collects a new dataset "UltraChat" that contains instructional conversations (Questions about the world, Creation and Writing, Assistance on Materials). The authors then use this dataset to fine-tune a new LLM based on LLaMA, and find that it achieves better performance than open LLMs not trained on such data.
The dataset is large and the reviewers judge it to be of high quality, and think that the semi-automatic data collection method is valuable for collecting similar such data in the future.

The reviewers' main concerns are about the cost of the dataset (and discussion of it; the authors address this to some extent in their rebuttal) and about a human evaluation for a subset of the data to obtain better ideas about dataset quality. For the rebuttal, the authors have conducted such a human evaluation at a small scale, so in my opinion the main concerns regarding the initial version of the paper can be addressed easily in the main paper; the one slighly lower score for this paper is based on a non-substantial review and can be disregarded.